# Improving Public Action to Mitigate River Flooding in Niamey (Niger)

**Saidou Oumarou Mahamane [1], Amadou Oumarou [1] and María José Piñeira Mantiñán [2,\*]**

[1] Department of Sociology and Anthropology, Abdou Moumouni University of Niamey, Niamey 10960, Niger; saidououmarou8@gmail.com (S.O.M.); mobangoula@gmail.com (A.O.)
[2] Department of Geography, University of Santiago de Compostela, 15782 Santiago de Compostela, Spain
[\*] Correspondence: mariajose.pineira@usc.es; Tel.: +34-881812626

**Abstract:** The purpose of this paper is to analyze the effects of repeated floods in the city of Niamey and the strategies developed by the state in terms of prevention, rehabilitation, and care for victims. Although numerous framework documents have been drafted in the last decade and urbanization in flood zones has been limited, the reality in Niamey shows that urban sprawl is increasing along the riverbank and the population facing vulnerable situations is growing. The inefficiency of state action determines that, on the one hand, it is non-governmental institutions that need to develop improvement plans—with the recovery of the city being dependent on the arrival of their funds and their distribution in those areas that donor countries consider most suitable—and on the other hand, it is the citizens—already vulnerable—who must find their own resilience mechanisms to try to survive the next flood. This study reveals that this diversity of players is involved in flood management through emergency relief, solidarity in rehousing, and providing support for living and non-living flood victims. Despite this mobilization, the actions undertaken are hampered by a number of constraints. Insufficient financial and human resources, a lack of foresight on the part of the authorities, and poorly coordinated actions are the main difficulties limiting the success of public action in the face of flooding. Through a qualitative approach, combining documentary research, direct observation and semi-structured interviews, we analyze the mobilization of actors around the management of the effects of floods in Niamey.

**Keywords:** river flooding; informal urban sprawl; public action; mobilization; Niamey; Niger

## 1. Introduction

Natural disasters are becoming more and more frequent around the world. According to climate change projections, this trend is expected to continue, leading to an increase in the frequency and instability of weather events [1]. Of these, floods are one of the most important natural hazards that sustainable development faces, reducing the assets of households, communities and societies through the destruction of standing crops, housing, infrastructure and buildings, not to mention the loss of life. In some cases, the effects of extreme floods are dramatic, not only at a household level but also for a country [2]. Natural disasters hit countries and households disproportionately. For example, between 1995 and 2014, 89 per cent of natural disaster victims lived in low-income countries, which were affected by only 26 per cent of these disasters [3]. Natural disasters are, therefore, considered one of the obstacles to development in these countries [4]. Since the 2000s, the number of floods has almost doubled around the world [5], as well as the number of people affected by floods and the resulting economic and financial losses [6]. Urban areas have been the most affected by increased flooding, which is becoming increasingly dangerous and costly to control because of the size of the exposed population. Floods affect small towns, commercial villages, medium-sized service centers, large cities and even metropolitan areas. Hence the urgency to consider flood risk in land-use planning

and management plans as well as in urban policies. In this sense, public actors need to be aware of the threats that may affect the urban environment in order to design measures to mitigate or solve the problems. Firstly, they need to understand that climate change will have a direct impact on flooding by (i) increasing the rate of sea level rise; (ii) changing local rainfall patterns, which could lead to more frequent riverine flooding, with higher levels and more intense flash floods; (iii) changing the frequency and duration of drought events, requiring groundwater extraction, which could lead to subsidence; and (iv) increasing the frequency of storms and storm surges. Secondly, they must exercise effective control over the uncontrolled growth of the city. There is no point in having legislation and planning if it is not enforced. Poorly planned and managed urbanization contributes to the risk of flooding. The massive influx of people into cities leads to an increase in paving and impervious surfaces; overcrowding; higher densities and congestion; and insufficient or ineffective drainage, sanitation infrastructure and solid waste management because they are too old or too poorly maintained [7–9]. In addition, much of this urban expansion is based on informal settlements, which are usually located in flood plains and flood-prone areas [6], and where the poor and general population lives, so the consequences of flooding in these areas are much greater and increase vulnerability levels [10].

In a context in which cities are becoming increasingly complex realities, from a physical, social, economic and environmental point of view, an integrated approach is needed to manage the risk of flooding and guarantee the inhabitants a better quality of life and more sustainable growth. This will require the implementation of structural and non-structural management measures that complement each other. The former should be aimed at reducing flood risk by (i) combining urban planning and management of drainage systems that must go hand in hand with urban growth and must have the capacity to evacuate water in the event of flooding [8,11], and (ii) combining hard engineering works—flood defenses, drainage channels—with more sustainable flood control measures such as wetlands and natural buffers. Non-structural measures should focus on early warning, should not require large investments and should be aimed at (i) emergency response planning and management (warning and evacuation); (ii) improving preparedness with awareness campaigns (maintenance of clean drains, better solid waste management); (iii) land use planning; and (iv) accelerating the pace of recovery and strengthening resilience (e.g., building back better). However, this is not an easy task. In general, the inhabitants prefer structural measures because they are "more comfortable" and do not involve any work, while with non-structural measures they must be more involved in maintenance and learning tasks.

In this context, the management of natural disasters, such as floods, is carried out by a wide spectrum of actors (state agents, communal authorities, NGO agents, technical service agents, and civil society actors) [12,13]. Thus, in these circumstances, even if these actors share the same objective "to relieve victimized populations", they do not all defend the same interests, nor are they guided by the same logic [14,15]. Generally speaking, in developing countries, the roles of state actors consist of framing interventions through the elaboration and adoption of strategic documents, providing victims with first aid through the reception of disaster victims in temporary shelters, and supporting them with food and non-food items [16]. However, countries in the South often face enormous difficulties in achieving successful flood management [17,18]. To some extent, these difficulties justify the intervention of humanitarian actors in disaster management. Indeed, since the declaration of the International Decade for Natural Disaster Reduction (IDNDR) 1990–1999, initiated by the United Nations, there has been a growing involvement of the latter in disaster management [19], whereby the mediators and brokers of aid and public action are mobilized, providing the link between these various arenas, while actively participating in them, and therefore, taking over from the state on a scale where the latter would be absent [15]. However, humanitarian actors, despite their apparent neutrality, carry a specific mandate for international governance consisting of program development, international project implementation, public education, advocacy, rights defense, and

volunteer programs. NGOs group together within national or international federations and associations to become interlocutors with governments, address common issues and concerns or act in concert when necessary [13,19].

In this article, we will focus on Niamey, one of the cities most affected by the floods of the Niger River. To begin, we will analyze the behavior of the river and the floods over the last few years, and investigate which urban sectors have been most affected. Then, we will analyze the roles of actors in the construction of public action in the face of the effects of flooding in Niamey. Specifically, this article aims to (1) describe the factors that lead to flooding in the city of Niamey, (2) explain the roles of the actors involved in management, and (3) analyze the constraints that reduce the effectiveness of the flood response.

## 2. Materials and Methods

This study adopts a qualitative socio-anthropological approach. The data collection is based on documentary research, observation and semi-directed interviews. The documentary research was performed at the documentation center of the Laboratory of Studies and Research on Social Dynamics and Local Development (LASDEL) and on the Internet. The exploitation of these documentary resources allowed us to gather important information that contributed to the analysis of the data. The observation focused mainly on the operations to identify victims and distribute aid to them. These observations contributed to the corpus of data.

The interviews consisted of semi-structured exchanges with various actors who make up the strategic groups. These are state actors, communal agents NGO agents, residents, and donors (Figure 1). In concrete terms, the interviews were conducted based on an interview guide. Discussions with the people we met revolved around themes relating to public action in the face of flooding. Specifically, they addressed the mobilization and participation of stakeholders in flood management, the types of action taken, the types of assistance received, the effectiveness of aid received in the face of the effects of flooding, and the difficulties that reduced the success of action taken. However, these themes are not systematically addressed to all categories of interviewees. The questions asked depended on the respondent's profile. The questions were open-ended, giving the respondent the latitude to express him/herself independently, while ensuring that the respondent focused on the priority issues of the study. The interviews lasted between 30 min and 1 h and 20 min. Data collection took approximately two months. A total of 38 interviews were conducted.

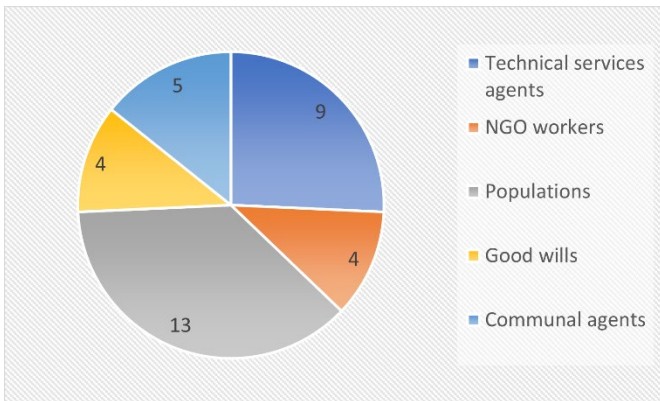

**Figure 1.** Distribution of interviewees according to profiles.

The notes taken at the end of these interviews were summarized, which made it possible to develop the corpus of data.

The data were analyzed using a thematic approach. The first step was to select the most important information from the mass of data, i.e., from the notebooks, and to classify it in a way that would make it easy to find [19]. The interviews were then categorized by

theme. Once the thematization had been completed, we moved on to contextualization, i.e., interpretation of the results.

## 3. Niamey: The Confrontation of Natural and Human Systems

Niamey, the capital of Niger, is on track to become the second fastest growing large city in Africa according to the 2010 *State African Cities Report* [20]. Since 1930, it has experienced significant population growth from 4000 inhabitants in 1930 to 1.3 million in 2021 and is projected to reach a population of 2.9 million by 2025. Therefore, the city will have experienced a growth rate of 525% in less than 100 years [21]. In part, this growth is due to the influx of immigrants from the countryside, who in the face of increasingly prolonged periods of drought, see the city as an opportunity to find work, have higher incomes and a higher standard of living [22]. Therefore, Niamey has experienced growth rates between 4.22% and 4.55% since 1990, and the growth rate reached 7.86% between 2010 and 2020 [23]. However, political instability (with several coups d'état) has contributed to the fact that the Niger government has had little power and has not paid attention to planning issues. Thus, since the development of the city masterplan in 1984, no planning tool has been designed, meaning regulations are seldom implemented. Furthermore, the city does not have a cohesive urban governance model, while the multiplicity of actors involved in urban management without coordination leads to urban duality in terms of investment and access to services and facilities. Thus, while some neighborhoods have garbage infrastructure, public lighting, or drainage channels, others lack the most basic services. As a result, a large part of the population is frustrated and refuses to pay taxes for services and infrastructure that they will not use, which in turn results in a reduced financial capacity for the state to improve them.

### 3.1. Niamey's Deregulated Urban Growth

Rapid population growth is pushing Niamey to an unprecedented size, but such uncontrolled urban expansion is linked to negative social and environmental issues. This is why several research papers have focused on modeling the urban growth of Niamey in order to understand the spatial and temporal urban dynamics. These studies provide tools for urban growth analysis for policy makers and city planners which help visualize different future scenarios as well as guide public intervention initiatives.

An assessment of urban growth and land use transition within the city was conducted using satellite imagery (Figure 2). Its results show that in the period from 1988 to 2030, the built-up areas in the city are expected to increase by 130% (going from 50.50 km$^2$ in 1988 to 116.14 km$^2$ in 2030), with an annual expansion rate of 3.09%. Furthermore, there will be a 125% increase in population density, going from 7870 persons/km$^2$ in 1988 to 17,769 persons/km$^2$ in 2030 [22]. The sprawl index or growth ratio indicates that there will be a three-fold increase in the population compared with areas that are being transitioned into urban use. On the other hand, taking into account that Niamey has a series of physical barriers that constrain its growth (the southwestern plateaus, ravines to the northeast and the green belt protection zone), urban expansion will take place in both the sectors located between the Niger River and the high plateaus of the southeast, as well as in the northwest and southwest zones, which have a more favorable physical environment (flatter topography) for the construction and establishment of urban infrastructure.

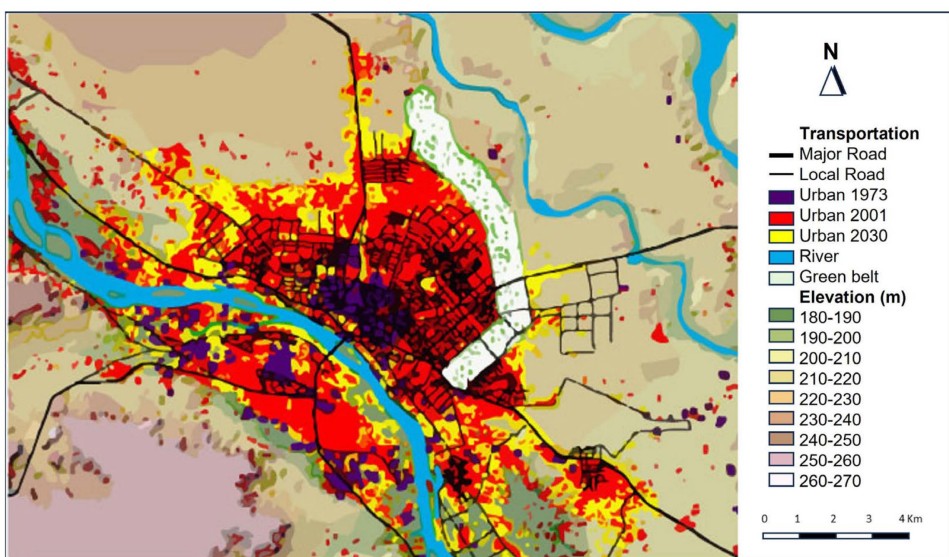

**Figure 2.** Predicted urban expansion for the city of Niamey for the period 1988–2030. Source: [22].

This physical and demographic growth has serious consequences for the environment by modifying natural systems, which increases the risk of floods and their impacts on the affected urban sectors. Urbanization has an impact on runoff because of the sealing of the soil caused by urbanization and the construction of infrastructure, meaning that the land has reduced capacity for rainwater infiltration. The most vulnerable area is on the right bank of the river [24], where there is already a proliferation of slums that are not equipped with urban sanitation facilities. In fact, the network of collectors dates back to 1981, and no more have been built since 1989, despite the fact that the city and the population continue to grow. This circumstance led NI-GETIP, with PRIU/World Bank financing, to build culverts in 1992 in some neighborhoods such as Banifandou and on the right bank [25]. However, if the construction of infrastructure (asphalt, bridges, railways) is carried out without considering certain parameters to protect the population from natural risks, as has happened in the past, the impact of floods will be enormous, considering that over the last few years there has been a massive felling of trees which serve to protect the soil and reduce runoff [25].

*3.2. Niger Floods in Niamey*

The Niger River is the third largest river in Africa (after the Nile and the Congo), both in terms of its length (4200 km) and the area of its basin (2,000,000 km$^2$), which covers the territories of 10 African countries in the following proportions: Algeria 3%, Benin 2%, Burkina Faso 4%, Cameroon 4%, Ivory Coast 1%, Guinea 6%, Mali 25%, Niger 22%, Nigeria 32%, and Chad 1% [26]. According to Babale's research, the Middle Niger regime is linked to rainfall [25]. The normalized flow index (EFI) at the Niamey station for the period 1950–2013, whose inter-annual flow is 877.39 m$^3$/s, has made it possible to distinguish three periods (Figure 3) [25]: (i) 1950/1951 to 1969/1970 had a flow of 1121.87 m$^3$/s and an inter-annual flow of 877.39 m$^3$/s; (ii) 1970/1971–1993/1994 was a period of drought, reaching a flow of 668.67 m$^3$/s, and representing a decrease of 27.78% of runoff in this period with respect to the series and a deficit of 40.39% compared with the previous period; and (iii) 1993/1994 to 2012/2013 was the wettest period with a flow of 894.70 m$^3$/s, with a substantial increase in runoff in contrast to the previous decade (1970/1980).

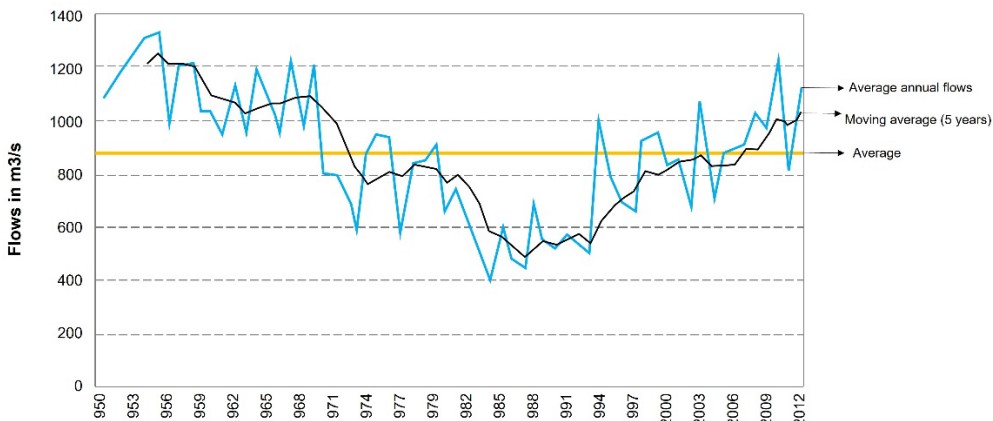

**Figure 3.** Average annual river flows in Niamey from 1950 to 2013. Source: [25].

This period of drought led to settlements in the supposedly inactive river branches and other flood-prone areas following the rapid growth of the city and the difficulty people faced in accessing housing in other parts of the city. The result was that such spaces were flooded on five occasions: 1998, 2010, 2012, 2013 and 2020. In 2012, the river, with a flow of 2492 m$^3$/s, reached a height of 6.18 m (Figure 4), almost a meter higher than the estimated warning level of 5.30, but this level was exceeded in 2020 when it reached 6.8 m.

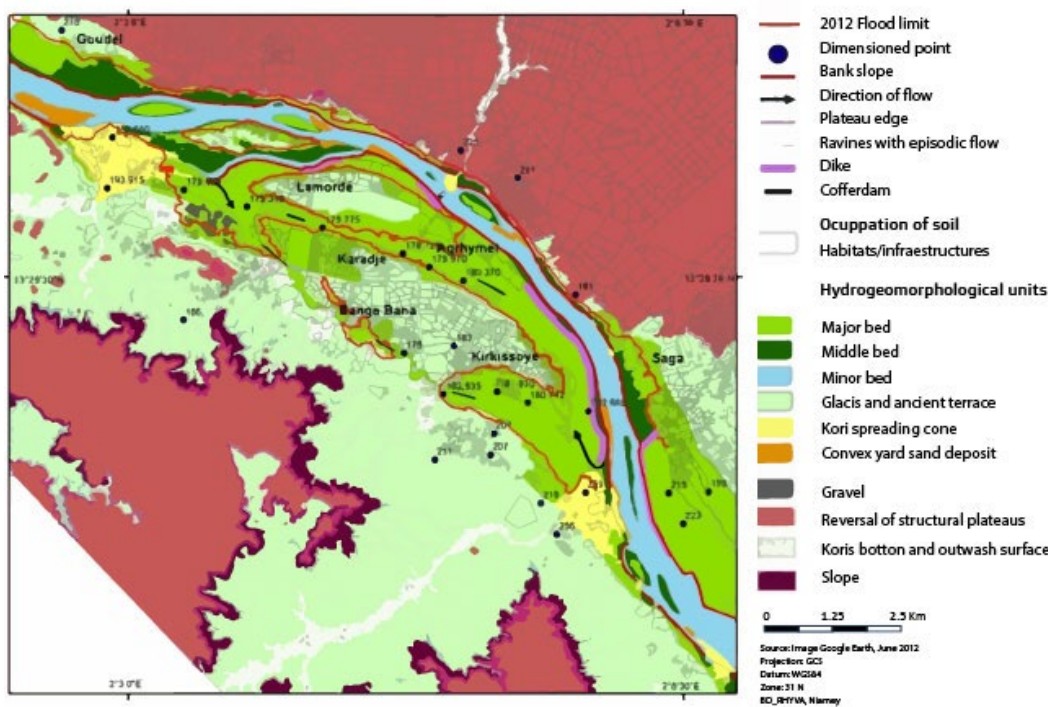

**Figure 4.** Hydrogeomorphology, land use of the alluvial plain of the Niger River in Niamey, and limit of the 2012 flood. Source [27].

Niamey is particularly affected by the overflowing of the Niamey River, which is reaching increasingly high levels of flow and causing recurrent flooding in built-up and cultivated areas, and thereby affecting thousands of people. The floods are becoming an unbearable risk for society, as well as for the survivors who must rebuild their homes and start again after each flood.

These floods are caused by both the Guinean and local rainfall. The first flood, which takes place between November and February, is caused by rainfall recorded in the mountainous massifs in Fouta Djalon, where the Niger River originates, and in Mali. This flood

is slow and has a relatively slow propagation time [28]. This flood, even if it is not perilous, constitutes a danger for the residents, especially those who occupy houses made of mud or straw huts, since it can last several days, or even weeks [24,29]. The second type of flood occurs between July and early September. It results from heavy rainfall recorded in the middle part of Niger, i.e., in the northern areas of Burkina Faso and in the extreme west of Niger, as well as rainfall recorded in Niamey and its surroundings, which contributes to the flows of numerous tributaries located downstream of the inner delta, including the tributaries upstream of the right bank of Niamey (Gorouol, Dargol, Sirba) that have their source in Burkina Faso [28,30]. The magnitude of this flood is a function of the amount of rainfall in the aforementioned localities. Moreover, studies explain that this runoff is increased by the destruction of the vegetation cover under the effect of droughts recorded during the decade from 1970 to 1980 and due to anthropic activity such as the exploitation of wood to satisfy the needs for energy and construction materials [30–34].

In Niamey, river floods have various devastating consequences, including the destruction of houses, crops, public infrastructure (schools, roads) and worse, the loss of human and animal life. Between 1998 and 2020, a total of 3,115,290 people and 7100 localities were affected by floods, with more than 225,000 houses destroyed, 205,000 ha of crops lost and some 46,540 head of livestock. The 2020 flood affected 52% of Niamey's localities, which recorded 4619 collapsed houses, 255 damaged huts, 1039 affected households and 72,638 people affected, according to estimates dated October 2020 [35]. On this occasion, the river engulfed the rice fields and completely flooded the districts on the right bank (Lamordé, Karadjé, Zarmaganday, Kirkissoye, Banga Bana, Nogar, Univerité), as well as the districts of Goudel and Saga (Figure 5), which are located on alluvial deposits. All of these districts are classified as high flood risk areas (Figure 6), but informal settlements continue to proliferate in these areas without the state taking appropriate measures to curb their growth or build infrastructure to mitigate the impact of flooding.

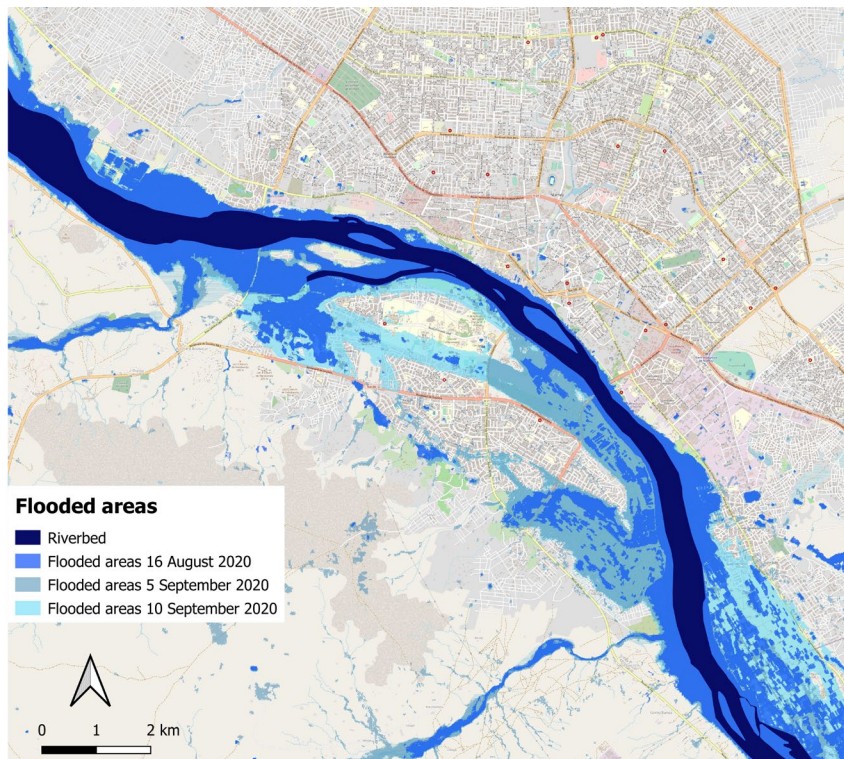

**Figure 5.** Map of flooded areas in 2020. Source: [36].

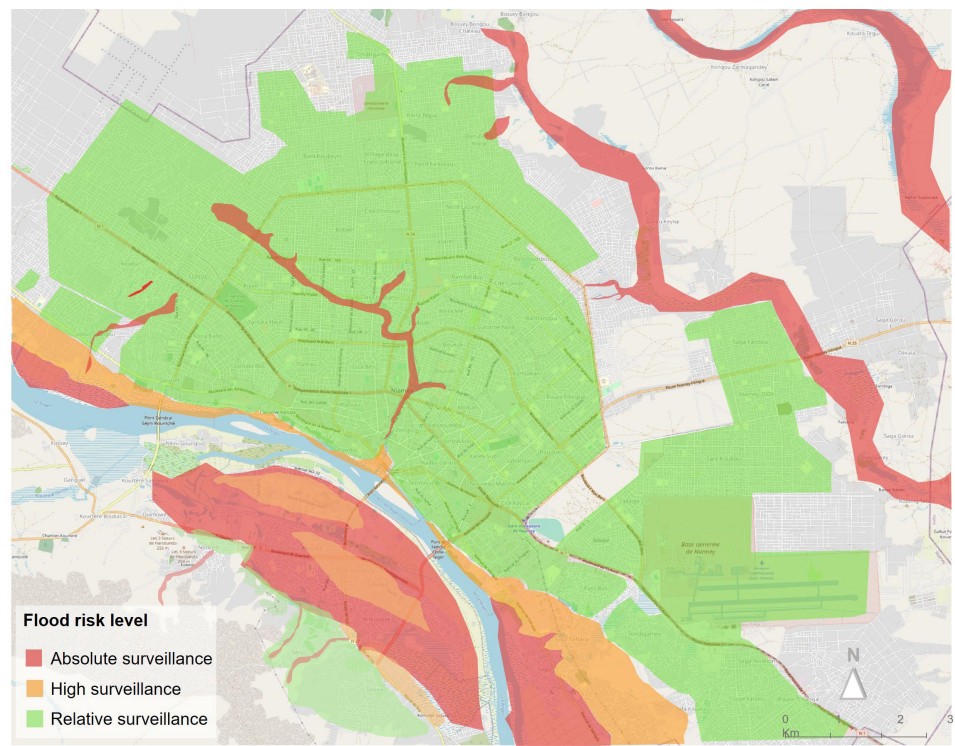

**Figure 6.** Map of flood risk level in Niamey. Source: [37].

## 4. Flood Management in Niamey: Structural and Non-Structural Measures

Floods, like other natural disasters, cause a significant number of people, groups and institutions to act, leading to their mobilization to help the victims cope. A mobilization can be defined as the process by which social actors work together to find solutions and act collectively in relation to what concerns them, which has effects not only at the structural level, but also at the personal level. This collective action is characterized by the sum of its two main components: a collective actor (us) and a collective action (common project) in the perspective of change [15,38].

Faced with the floods in Niamey, many actors mobilized to help the victims, with a distinction being made between those of a governmental and non-governmental nature (Table 1).

**Table 1.** Overview of governmental and non-governmental actions.

| Nature of Actions | Actor | Measure Type | Actions | Weaknesses |
|---|---|---|---|---|
| Governmental | State | Prevention, reconstruction | Flood management plans | They are mostly non-binding documents. When they are binding, they do not respect the criteria for land use and spatial planning. |
| | | Mitigation | Assistance to victims: Census. | Insufficient human resources Lack of anticipation The behavior of the inhabitants |
| | | Mitigation | Agreements with other institutions | |
| Non-governmental | Political parties | Prevention, reconstruction | Political programs | Political competition and appropriation of financial resources |
| | NGO | Mitigation | Victim care and rehabilitation actions | Action when disaster strikes, not preventive action. |
| | Well-intentioned individuals | | | |

*4.1. State Mobilization: Ineffective and Late Actions*

There is no doubt that the state is the main actor in flood management. This management is organized around three organizations: the Regional Device for Food Crisis Prevention and Management (DRPGCA), which is in charge of receiving donations and distributing them to the victims; the General Directorate of Civil Protection (DGPC); and the HFA/GC. Between them, they promote the following:

1. Flood management plans, emergency organization plans, community rescue plans, and data collection related to the disaster through censuses of victims and those affected.
2. Actions to alleviate the suffering of the population, through rescue operations and assistance to the victims.

Among the plans that have been approved since 2017, *The Communal or Intercommunal Safeguard Plan* [39,40], which is aimed at the prevention of natural risks and the definition of risk areas to organize protection measures for them, stands out. In 2018, a flood risk analysis of the communities of the Dosso and Niger region was carried out to ensure food security, and in 2020, three key documents were produced: The *Flood Contingency Plan*; the *Niger Flood Response Plan*; and the *Support Plan for Vulnerable Populations* [41–43]. These documents highlight the administration's concern about the need to contribute to strengthening the capacities of state structures so that they are equipped for greater efficiency in providing a humanitarian response [16] and to have a better understanding of the degree of vulnerability to which the affected population is subjected.

More precisely, in order to obtain greater knowledge about this last aspect, the state compiles a census of victims, the second most important activity that the state must carry out after providing temporary shelter to flood-affected victims. A census is carried out by the fire department and is considered crucial for collecting information on the number of people affected, the houses that have collapsed, the area of crops damaged, etc.

Census operations usually begin within 24 h of flooding. They are coordinated by the General Directorate of Civil Protection (DGPC) and supported by the Niger Red Cross. Since 2012, the DGPC is the only state structure authorized to process the information as problems have previously been observed.

"After the 2012 floods, several assessments announced divergent results on the number of victims. Each actor made its own assessments. This raised doubts about the reliability of the data, which led to some mistrust on the part of the authorities. In these operations, people brought out their political colors. That is why, since then, the government has decided that only the protection and the red cross are authorized to carry out assessments." (DGPC official, interview on 4 April 2019 in Niamey).

However, in addition to the census conducted by the DGPC, in some cases, other censuses are conducted by disaster victims and other humanitarian NGOs. In practice, census operations in the case of floods are hampered by three main constraints:

- Insufficient human resources: The assessment of floods, especially when they are large-scale, requires a significant human mobilization. The floods in Niamey have the particularity of occurring in several communal districts as well as in several neighborhoods of the same communal district, which requires the deployment of agents in several areas at the same time, thus spreading personnel thinly on the ground. The DGPC and the Red Cross do not have sufficient personnel for this type of operation. In addition, the DGPC suffers from insufficient human resources for its implementation as planned at the department and commune level.

"Firefighters alone cannot conduct the census. They do not have enough personnel to do it. That is why they rely on the support of Red Cross agents in the census operations." (Government official interviewed on 1 July 2020, in Niamey).

During floods, many people, especially local disaster delegates, are responsible for taking a census. This census usually takes place three or four days after the floods occur. It is conducted after the failure of operations carried out by the DGPC and the Red Cross. In some cases, people have been known to photocopy forms that they fill out themselves and give them to their delegates, leading to disputes among the victims and even violence.

"Several censuses have been conducted. There are rejected censuses. There was a first census. People contested this census. There was even a fight. The list was broken. Then, the delegates took the census of the members of their classes." (Resident of the Neino Goungou neighborhood, interviewed on 15 September 2020 in Lamordé).

Faced with insufficient staff, the municipal authorities sometimes carry out a census of the victims by authorizing some of their agents to do so.

"We carried out the census in collaboration with the neighborhood chiefs. The census was done in schools. We did not manage to take a census of all the victims. That is why, afterwards, people bring in their lists. I pass the list on to the partners who ask for it." (Head of the Humanitarian Affairs Department, Niamey 4 communal district, interview on 12 January 2021 in Niamey).

In some cases, the census carried out by the DGPC is questioned by the disaster victims, which leads them to carry out other censuses, as this delegate explains in these terms:

"We had done our own census of the evacuees who are in this facility. Following the cash support that care gave, we received complaints that several people who received this money are not disaster victims. So, we informed the mayor of the 5th district. He said to draw up a list. Following this, the delegates of the different classes drew up their lists which we sent to the mayor." (Delegate General of the CES Lamordé site, interview on 15 September 2020 at CES Lamordé).

In summary, in terms of assessment, and despite the authorities' willingness to organize operations around the DGPC and the Red Cross, several actors are involved in the production of flood figures. The lack of human resources and the politicization of flood management are two major obstacles to the effective implementation of this assistance. In the end, we end up with contradictory, biased and therefore, unreliable figures for good flood management.

- Lack of anticipation: Although the plans should identify the risks to which the population is exposed and outline interventions, the truth is that there is a lack of training in terms of knowledge of the terrain and the mechanisms/skills required for intervention. In addition, the people who carry out the censuses do not have adequate training: We cannot take people who are not even from Niamey, who do not know the neighborhoods, and expect them to take a good census. This census must be carried out with the involvement of all the actors, precisely the people who are from the neighborhood, by associating the neighborhood chiefs, the communal authorities and the sons of the neighborhood as census takers. (An agent from the Niamey 5 communal district, informal interview, 10 October 2020).

On the other hand, it has been observed that the census conducted in 2020 did not fully include all the victims in the city of Niamey, for example, the victims of the airport and Saga neighborhoods in the Niamey 4 communal district, and those of the SONUCI neighborhood in the Niamey I communal district, as shown in Table 2.

**Table 2.** Communal districts affected by floods in 2020.

| Region | Depart. | Commun. | Neighborhoods | Households | Total Households | | |
|---|---|---|---|---|---|---|---|
| | | | | | Commun. | Depart. | Region |
| Niamey | Niamey | ACN1 | Gabgoura | 65 | 89 | 3146 | 3146 |
| | | | Yantala Corniche | 8 | | | |
| | | | Gorou Banda | 1 | | | |
| | | | Lossogoungou | 15 | | | |
| | | ACN3 | | 8 | 8 | | |
| | | ACN5 | Kirkissoye 1 et 2 | 947 | 3049 | | |
| | | | Nourou Salam-Aimé Césair | 68 | | | |
| | | | Kirkissoye 3 | 83 | | | |
| | | | Gaweye 1 | 72 | | | |
| | | | Gaweye 2 | 185 | | | |
| | | | Néné Koukou | 21 | | | |
| | | | Saguiya Rizière | 98 | | | |
| | | | Lamordé 1 | 561 | | | |
| | | | Pont Kennedy | 51 | | | |
| | | | Kenigoungou | 51 | | | |
| | | | Tassi Konou | 423 | | | |
| | | | Lamordé 2 et CEG Lamordé | 489 | | | |

Source: [42].

In short, whatever procedure is adopted, it is difficult to arrive at exact figures, due to the diverse obstacles faced. In the case of flood impact management, the figures become political and economic issues.

"The figures do not reflect the reality of those actually flooded. The numbers affected are exorbitant. The method of counting households has limitations. Several members of the same household can present themselves as separate households. All that is needed is an identity document. People have not understood the purpose of the census. They think that by declaring that they have lost a lot, they will receive a lot of help." (Resident of Lamordé, victim of the catastrophe, interview on 14 September 2020 in Niamey).

The discrepancies observed refer to different competing or overlapping needs to make the event intelligible, to offer legitimacy to the actors, to compensate for reconstruction, to access aid, to ensure the sustainability of human resources and investments, and to prioritize decisions. In these circumstances, the symbolic power of the language of numbers is at the heart of sociopolitical issues [44].

- The behavior of the inhabitants: The impact of floods is greater on low-income households. Being registered on a list of disaster victims is a guarantee that "anonymous" victims will be able to access assistance services. However, the system becomes corrupt when non-victims develop strategies to register and even benefit from the complicity of the disaster victims.

"Many people who are not victims of the catastrophe have registered. That is why they have managed to benefit from the aid intended for the victims of the catastrophe. We are afraid to denounce them. If you denounce someone, they call you a hypocrite. At one point, we had removed the names of all those who were not victims of the disaster, but the head of the district intervened to tell them to leave them alone, that it was their chance." (General delegate of the CES Lamordé site, interview on 15 September 2020 at the CES Lamordé).

"The census of the victims is quite late. The firemen started, they didn't finish when they suspended. During this census, people come with the cards of other people who are not victims to be counted. Those of us who denounce this are frowned upon. It is usually the women who do it." (Councilor of the communal district Niamey 5, interview on 16 September 2020 at the CEG 9 of Niamey).

This is compounded by the fact that delegates take advantage of their status to register non-victims.

"The delegates are not honest. They take advantage of this role to include their relatives in the lists of victims. This is the example of CEG 09. They brought me a list with more than 1700 victims. No one can believe these figures." (Head of the Department of School Affairs, 5th arrondissement, informal interview, 10 October 2020).

"We had made a census of all the victims in each class. The classes consist of between 20 and 35 people. This census has its limitations. The main limitation is that the delegates write down the names of people who are not victims of the catastrophe. Even if there is a distribution, people from here call people from the village to come and take advantage of the aid." (Informal interview with the delegate of Nordiré, interview on 19 September 2020 in Nordiré).

The state of the land: Floods wash away homes and the space is occupied by water for several days, forcing people to abandon their homes and seek shelter. The ideal scenario would be to conduct the census in the home, but in such circumstances the enumerators must conduct the interviews in schools or in the street (Figure 7), making it difficult to control whether they are surveying the entire population or not.

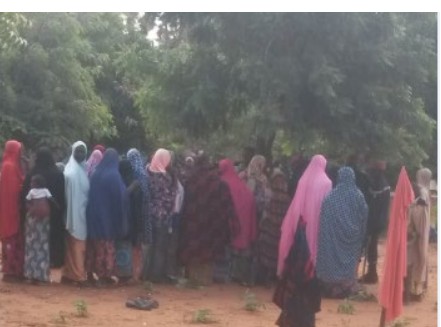 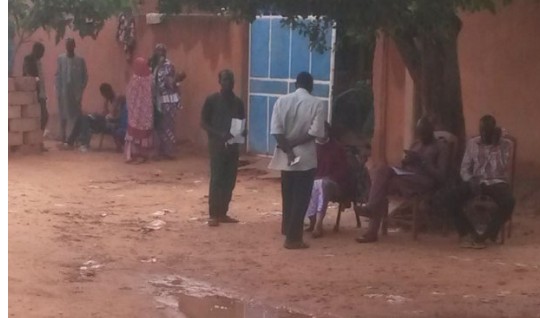

**Figure 7.** Census operation in a school and a street in the Niamey 5 communal district. Sources: Oumarou Mahamane Saidou, 27 August 2020.

With regard to the rescue operations and assistance to the victims, it should be noted that the state employs a variety of strategies to request international aid in different geopolitical contexts [45] and thus raise funds for reconstruction. These strategies include the signing of agreements between ministries and institutions such as those of the United Nations system, the use of the State–Donor Framework Agreement, the World Bank's immediate response mechanism and the appeal for national solidarity (Table 3).

**Table 3.** Resource mobilization strategies.

| Strategy | Objective | Weakness |
|---|---|---|
| Signing of agreements | Collaboration of a technical and financial partner. The planned activities are part of the partner's program. | Obtaining funding depends on the competence and influence that the ministry or the agent of the requesting service has over the agents of the partner institution. Aid is obtained at the cost of submission to the objectives and priorities of the donors, which are formulated in a somewhat direct manner [45]. This strategy is often mobilized for advocacy, training and post-disaster recovery activities. The actions carried out on the ground are not very visible. |
| The use of the donor–state consultation framework | Fundraising strategy used by the DNPGCA. The main objective of this mobilization is to supply the national food security stocks housed at the Office des Produits Véveres du Niger (OPVN). The rationale of this framework is to mobilize resources to address food insecurity. | This framework suffers from weak of financial support from partners. Donors are increasingly turning to international NGOs to the detriment of state structures. The "standard" procedures and actions of international agencies lead to costly "duplication" in terms of time and resources and destabilize national technical and political bodies, which feel challenged, undervalued or ignored [46]. Donor representatives are deprived of much of their decision-making power to the benefit of headquarters located in the North. The headquarters are highly sensitive to media pressures on the one hand, and unfamiliar with the local situation on the other. They manage their interventions in Niger directly from offices in New York, Paris, Washington or Dakar. |
| Rapid response mechanisms | The objective is to provide immediate basic assistance to populations affected by armed conflicts, natural disasters or epidemics. | Despite the delay in the release of funds, it allows the state to have funds available for the implementation of its response plan. If these funds are not managed rationally, the objectives set will be far from being achieved. |
| Call for national solidarity | This strategy usually takes place after the authorities' field visits. The objective of this mobilization is to inform the public of the extent of the floods through images and videos, while conveying the government's call for national solidarity with flood victims. | Most often, this appeal is followed by contributions from civil society actors, companies or even individuals, other countries or international institutions. |

However, despite this diversity of resource mobilization strategies, government efforts are not always commensurate with the needs of flood victims.

*4.2. Non-State Intervention: Support for Public Action and the Challenge of Coordination*

Non-governmental organizations have played an important role in crisis management for several decades [4]. In an international context of increasing disasters and the retreat of state authority, actors such as NGOs, political parties and well-intentioned individuals are increasingly involved in disaster relief, recovery and reconstruction [47,48].

Because of their speed of intervention, expertise and ability to help victims, NGOs have quickly emerged as alternative models to the state [49], and they all operate in the name of humanitarian action. In Niger, as in neighboring countries, NGOs have multiplied since the 1990s, with the confluence of the political opening allowed by democratic transitions, the evolution of aid policies that have valorized "civil society" and their role in development, and the employment crisis caused by structural adjustment policies that have led former laid off civil servants and young graduates to try their luck in the NGO sector [45]. While their aid includes food, blankets, mats, buckets, soap and money (Figure 8), sometimes the support of some NGOs goes beyond emergency aid and continues with rehousing operations for disaster victims, as was the case of Qatar Charity after the 2012 floods.

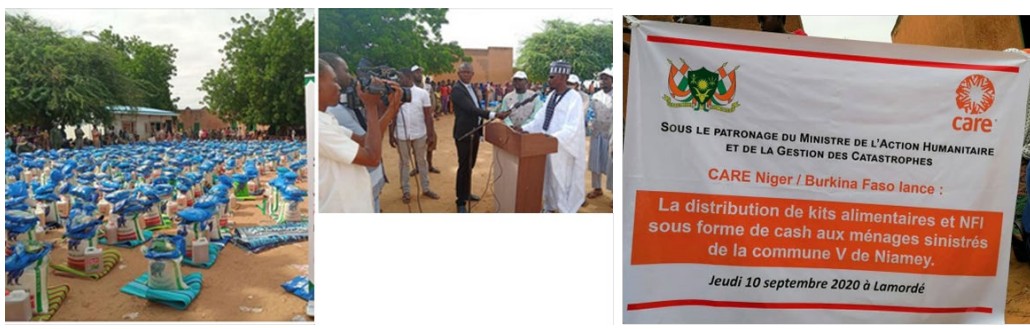

**Figure 8.** Work of the NGOs Karkara (**left** and **center**) and Care (**right**).

As far as political parties are concerned, in the context of political pluralism marked by a permanent struggle for political power, helping the population becomes, in addition to an act of humanism, a strategy for mobilizing the electorate. Thus, parties intervene in local territorial areas—which they perceive as spaces of political competition and appropriation of financial resources—to provide aid for the construction or repair of infrastructure (religious buildings, village water supplies, school buildings or health facilities), the payment of taxes to the state or the commune or of co-payments required by projects, and food aid in case of food shortages [50]. To make the mobilization visible, these actors use various channels for media coverage, including radio, television, Facebook pages, WhatsApp groups etc. In addition, it should be noted that in some cases, the mobilization of these political actors is initiated by the disaster victims themselves. This is part of the strategy developed by the latter to multiply donors.

Finally, we must refer to the generosity of anonymous persons, associations, training centers, traders' groups, etc. They are provided, not to fulfil obligations (like state structures) or to carry out an action to justify its expenses and seek other funding (NGOs), but to fulfil a moral and religious duty. The orientation of these interventions is less influenced by politicians, the local executive and even less by administrative agents. This aid is most often offered anonymously, without media interference.

## 5. Discussion

In most African countries, urban water management is based on outdated practices and professionals receive very little continuous training. It is striking that according to the research by Lumbroso [10], out of 54 sovereign states in Africa, only in 7 do the stakeholders of the territory perceive flood risk assessments to be effective. This is a worrying fact, because while it is true that many cities develop flood management plans, they are not made visible as effective due to the limitations (in terms of personnel, finances and management) of the administrations.

In the case of Niamey, we have found that flooding is recurrent, and although there is a concern on the part of the state to analyze the risks and understand the vulnerability of urbanized spaces, there is a lack of effectiveness in the prevention of the impacts of floods, as well as in the management and coordination in the event of a disaster.

In terms of prevention, we have seen that for years, multiple plans have been published that delimit areas at risk of floods and establish guidelines to mitigate their effects, while in terms of urban planning, it has been determined that flood-prone areas are those declared undevelopable by the city's *Urban Development and Management Master Plan* (SDAU). However, the reality is that the riverbanks continue to be urbanized, especially the right bank, which is notably the most vulnerable. This circumstance, in turn, translates into a greater flooding impact due to primary traffic lanes built on the embankment and forming a barrier. The drains are undersized due to the underestimation of the contribution of unpaved surfaces to runoff (poorly adapted models), as well as errors in the evolution of urbanization at the end of the structure's useful life. There are also impediments to runoff at the outlet and concession walls built over natural water flow pathways, etc.

The most serious aspect is that the general population, despite awareness of the risk and the cyclical nature of floods, still yearn to occupy these spaces, since their objective is to find a site on which to build a shelter [50]. This circumstance is not limited to the traditional and socially more vulnerable districts, but can also be observed in districts such as Koira-Kano, a symbol of the economic success of the 1980s, where luxury housing abounds. The occupation of this area, despite being in a flood zone, is due to the desire to access a plot of official land—regardless of geographical location or topography—as supply has always been lower than demand. The most striking thing is that although the plots must have a maintenance service, the municipality is not in charge of this task, so it is the buyers who, according to their means, carry out the maintenance. It is, therefore, not surprising that the wealthy, who are usually the first to develop their plots, do so to the detriment of their neighbors.

In the area of management, a lack of professionals trained to intervene in the territory has been observed, and the lack of resources does not help. In this sense, the administration is subordinated to the arrival of economic aid and its distribution by the different territorial areas by various partners.

Added to this is the lack of sensitivity on the part of the population to abide by the guidelines established by the administration in terms of risks, learning and doing things better. Flood survivors prefer non-structural measures (economic aid and reconstruction) to others that imply having to build in other areas, with other materials, keep watercourses clean, etc.

Regarding flood management, the study notes that, despite the diversity of players mobilized for this purpose, numerous constraints considerably reduce the success of various interventions. On the one hand, the state, as the main player, is faced with insufficient financial and human resources to effectively carry out all operations. In this context, state agents (technical services) behave like predators [45], claiming resources and roles for themselves and thereby taking their partners' resources [50]. Under these circumstances, we witness institutional pluralism in flood management, which often leads to conflicts of interest between state actors and a lack of coordinated intervention, constituting a major obstacle to effective flood management [51]. On the other hand, despite their apparent contribution in terms of financial resources and the technical skills of their staff, the way in which technical and financial partners provide "aid" is becoming a constraint in flood management.

Indeed, each actor is obsessed with implementing his or her own strategies with a view to appropriating the capital valued in the field, including economic capital (financing), social capital (functional relationships, actor networks), or symbolic capital (expertise, training, skills, experience or surplus value added to actor characteristics such as social position, status or profession) [51–53]. Thus, beyond the emergence of non-state and private actors in the governance of service production, the state is also in competition and short-circuited in its daily intervention by parallel bureaucracies implanted by a highly complex and dynamic network of international actors [13]. In this case, we are witnessing duplication and poorly distributed aid in the field of flood management.

However, it is worth highlighting the actions that some African cities are promoting to manage floods [54]. In Accra, for example, vegetation is being planted on slopes to prevent erosion and around water bodies that serve as drainage systems for communities. Also, they are building culverts and stormwater drains in flood hotspots, educating citizens on responsible waste disposal to prevent clogging of drains, and demolishing houses built on waterways. In Cape Town, there are actions aimed at the restoration of sand dunes; protection of kelp beds and Ramsar-designated wetlands to reduce coastal flooding and dissipate tidal energy; and construction of sea walls along the coastline and gravel platforms under residential dwellings to reduce flood exposure. In the case of Durban, they are cleaning waterways and restoring river areas through different projects that help people in local communities find a job; and installing groynes to redirect water flow in streams and rivers. In Nairobi, actions are focused on the incorporation of green infrastructure; evictions of illegal settlements from riparian areas and flood plains to restore natural

drainage pathways; the restoration of wetlands to improve flood water storage capacity; and the improvement of drainage in and around houses. Finally, in Mombasa, greening programs are being integrated, particularly in transport infrastructure and public spaces; and trunk drainage systems are being improved.

## 6. Conclusions

In Niamey assistance to flood victims is mainly the work of the state, which intervenes through the provision of temporary shelters, notably classrooms, and the distribution of food and non-food items. However, in its efforts to assist the population, the state is faced with insufficient financial resources. This forces it to develop strategies to mobilize these resources by seeking support from technical and financial partners and calling for national solidarity.

Given the inability of the state to provide sufficient assistance to flood victims, other actors such as NGOs and other compassionate individuals provide support to alleviate the suffering of the victims. Their support also consists of food and non-food items, and in rare cases, money. Most of the time, the support of these actors is ad hoc and does not allow for an effective management of the needs of the victims.

In summary, public action to deal with the effects of floods suffers from many short-comings. These are mainly the low investment from the state, the lack of preparation before the occurrence of floods, and the poor coordination of interventions on the ground. These actions could be improved if the state invested more by creating a fund dedicated to flood management and by setting up an effective coordination mechanism. It should also develop new relocation strategies for the populations affected by floods since the current strategies are not efficient, resulting in populations returning to flooded areas near the river. As a result, they suffer serious situations of vulnerability due to the lack of minimum services such as a water supply and have to rely on individual resilience to mitigate the danger in the event of a new flood.

Therefore, it is urgent that the administrations begin to manage the planning of their cities and the management of their territory more efficiently. On the one hand, it is necessary for the different existing plans to become binding documents for land use planning and the management of urbanization and infrastructure in areas at risk of flooding, On the other hand, the state must offer support to the cities suffering from floods by implementing the recommendations of the Yokohama Conference in 1994 and the Sendai Conference in 2015 [55,56] that states invest in disaster risk management.

**Author Contributions:** Conceptualization, M.J.P.M.; methodology, S.O.M. and A.O.; formal analysis, M.J.P.M. and S.O.M.; investigation, S.O.M. and A.O.; writing—original draft preparation, M.J.P.M. and S.O.M.; writing—review and editing, M.J.P.M. and S.O.M.; supervision, M.J.P.M. All authors have read and agreed to the published version of the manuscript.

**Funding:** This research received no external funding.

**Data Availability Statement:** Not applicable.

**Conflicts of Interest:** The authors declare no conflict of interest.

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
