# Peer review of "Improving Public Action to Mitigate River Flooding in Niamey (Niger)"

_land, doi:10.3390/land12081523_

Round 1
Reviewer 1 Report
Your paper is well structured, organized, and the research objectives are clear. In general, I find that this paper will be a strong addition to the research in political actualization, especially as it engages a socio-ecological approach for mitigating and adapting to semi-natural disasters.
Below is a series of minor recommendations for improving the quality of the paper presentation:
+ The Abstract should include the findings of the work, specifically related to - insufficient human resources, lack of anticipation, behavior (education) of the population - and the need to support non-state intervention.
+ In the Introduction, lines 82-86 provide a one sentence paragraph beginning with, "Therefore, it is urgent...." that should be moved to the conclusion of the paper. In a research paper like this, advocacy for a particular course of action should only emerge with the findings.
+ In the Methods section, lines 115-118 discussing the "corpus of data" needs to be more explicit. While "triangulating the data" is important articulating how this triangulation was conducted (word-based, theme-based, etc.) is necessary. Just provide more detail on how this was conducted.
+ In section 4.1, for your three primary findings for non-structural measures listed as bullets could be incorporated into a table near the front of the section, similar to Table 3. This would make it easier for the reader to better understand your explicit findings.
The paper is well written in clear English though it should be extensively reviewed for grammatical errors and spelling.
Explicit notes:
+ The Title for section 4.1 is in French. It should be translated to English.
+ The majority of the References are listed in French. These should be translated to English, where appropriate.
Author Response
Dear reviewer
Many thanks for your comments. I send the answer in the atached file.
Kind regards
María José

Reviewer 2 Report
Dear Authors,
After carefully reading your manuscript, I suggest the following improvements:
1. Change table one to a figure of circle sectors.
2. Please review and correct any units that have exponents, such as areas or flows. (for example those found in lines: 149, 151, 181, 186, 187, etc.).
3. Translate the legend included on the map of figure 3 from French to English.
4. Translate the title on line 255 from Spanish to English.
5.Undoubtedly, the description made is exceptional. However, I believe that a proposal based on a particular methodology should be included in the discussion section to improve the situation and suggest a sequence of stages to build a strategic plan that leads to greater system resilience.
Suggestions have been made in previous comments.
Author Response

(The authors gave the same response as above.)

Reviewer 3 Report
Thank you for the paper entitled "Construction of public action against river floods in Niamey (Niger)", submitted to the special issue of the Land journal. The subject addressed is generally within the scope of the journal and the special issue. The paper was quite interesting to me, as it presents the strategies and mechanisms of public actions – prevention and rehabilitation – against floods in one of the developing African county. Particularly the identification of problems with such prevention and rehabilitation of livestock and victims is important issue, taking account the economic situation in Niger. So, despite the simple research methods and qualitative type of the analysis, the significance of the research leads me to give positive feedback to the authors. However, before publication in Land I would have to request a few revision. My general recommendations are listed below:
- In the abstract, some major conclusions form the conducted qualitative analysis should be provided.
- The introduction section should provide more detailed insight into recent research about strategies of public and non-government organizations against the floods. Moreover, the objectives of the paper were not clearly stated; please provide specific aims of the work.
- More information about methods would be welcome; it is not clear to me what is the manual analysis of the data? (line 116). Despite the fact that this is a qualitative work, the methods should be described precisely and concisely.
- Section 3.1 – is it possible to provide detailed data about changes of impervious area in the Niamey city during the last year?
- Figure 2 – please provide the legend of the chart in different way, currently it is not readable.
- The title of the 4.1 section is not given in English.
- Discussion section is too short and superficial and it should be definitely strengthened with additional references. Finally, in my opinion too little attention has been given to flood strategies in different regions of the Africa, which will be interesting to the international reader. Maybe the problems with the prevention and rehabilitation are similar in whole sets of countries, depending on their socio-economic status? I suggest to expand this section.
I am not a native speaker but in my opinion English of the manuscript should be a little improved in some parts to be more sophisticated.
Author Response

(The authors gave the same response as above.)

Reviewer 4 Report
The paper Construction of public action against river floods in Niamey (Niger) presents a well-documented study on the public action measures taken to limit the consequences of floods in the city of Niamey from Niger. It is also well structured and written in a good English language.
The paper has 15,5 pages of which about 1,5 for the Introduction, 0,5 pages for the Materials and methods, 4.5 pages for integrating the natural and human background in the area 5,5 pages for presenting the flood management measures/actions 1 page for Discussion and 0,5 page for Conclusions. References are cited mostly from national/international projects, thesis, conferences or books, only about 5/47 references are from papers published in scientific journals.
There are some scientific issues (A) regarding the method and results and also some formal corrections (B) to be made before publishing the paper.
A. Scientific issues:
1. In terms of the methods used, the authors could have given details on the interviews guide, or presented the questions that were asked to collect the data so that the results of the study were clearer.
2. The flood management methods are not very clearly presented. They could be classified according to the disaster cycle as measures taken after a previous flood – such as emergency and reconstruction (clearly presented) and measures taken before the next flood, such as prevention and mitigation - of structural and nonstructural type (not clearly explained, although the name of chapter 4 includes them).
B. Formal issues
1. Table 2 column between Commun. and Household should have maybe Name in the head of the table
2. Please translate title 4.1 in English
3. Please translate Fig. 5 legend in English
4. Remove double phrase from R 340-351
5. Ref. 6 and 10 are the same?
Author Response

(The authors gave the same response as above.)

Round 2
Reviewer 3 Report
The manuscript has been improved and the authors have tried to do their best. I would recommend the acceptance of the manuscript for publication in Land journal.